# Examining the Effects of Mirror Therapy on Psychological Readiness and Perception of Pain in ACL-Injured Female Football Players

**DOI:** 10.3390/jfmk7040113

**Published:** 2022-12-14

**Authors:** Tiziana D’Isanto, Francesca D’Elia, Giovanni Esposito, Gaetano Altavilla, Gaetano Raiola

**Affiliations:** 1Department of Human, Philosophical and Education Sciences, University of Salerno, 84084 Fisciano, Italy; 2Department of Political and Social Studies, University of Salerno, 84084 Fisciano, Italy

**Keywords:** cognitive approach, LCA, football, psychological readiness, rehabilitation

## Abstract

Virtual reality-guided imagery (VRGI) and mirror therapy (MT) have been used in isolation to treat patients suffering from different injuries. However, no attempts have been made to understand the effects of combined VRGI and MT added to conventional physical therapy, and no information exists regarding perceptual responses to these rehabilitation strategies in female football players. Thus, this study aimed to examine the effect of MT added to conventional rehabilitation on psychological readiness and perception of pain in ACL-injured female football players. Thirty ACL-injured female football players competing in the 2nd and 3rd Italian tier who underwent an ACL rehabilitation program from the same clinic participated in this study. Players were randomly distributed in an MT group (n = 15) and a CON group (n = 15). All participants reported their perception of pain on a VAS before and after the interventions and their psychological readiness to return to sport after ACL injury and reconstruction surgery on the ACL-RSI scale after the intervention. An independent-sample *t*-test was performed to assess between-group differences in post-intervention ACL-RSI, and a further independent-sample *t*-test to assume non-significant differences between VAS values before the intervention. A two-way repeated-measures analysis of variance was used to test the null hypothesis of no different change in VAS over time between groups. After the intervention, the MT group perceived largely greater psychological readiness (*p* < 0.01). MT and CON groups experienced a large reduction in VAS after the intervention (*p* < 0.001). However, a small time × group interaction was observed (*p* = 0.023). MT reported a greater perception of the psychological readiness of the soccer players and a lower perception of pain than those who performed conventional therapy.

## 1. Introduction

The knee, with special emphasis on the anterior cruciate ligament (ACL), is often cited as the most common injury site in female football players [1,2,3,4]. Females are two to eight times more likely to be injured than their male counterparts, probably because male and female neuromuscular patterns diverge during and after puberty [5,6,7]. Biomechanical factors related to ACL injury risk factors have been extensively documented in female football players [8], underlining slower and less frequent contractions and increased knee valgus moments during landing compared to male players [9]. Accordingly, return-to-play strategies following ACL injuries in female football players have been focused on biomechanical-related performance variables (e.g., derived from a countermovement jump) [10]. However, no information is available regarding perceptual variables during the return to play in female football players.

Psychological responses to specific rehabilitation programs have been identified as a potentially modifiable factor associated with knee function [11,12,13]. Indeed, attempts to understand psychological factors associated with ACL injury rehabilitation have significantly increased. In particular, greater psychological readiness to return to sport and lower fear of re-injury favor returning to the pre-injury level sport after ACL reconstruction [14]. Psychological readiness to return to practice and competition is a combination of self-confidence and remaining uninjured, as well as the lack of fear and anxiety [15]. In this context, the Anterior Cruciate Ligament Return to Sport After Injury (ACL-RSI) questionnaire is widely accepted among clinicians and practitioners involved in rehabilitation settings to assess psychological readiness [16]. A previous study conducted on athletes who underwent ACL reconstruction revealed that using the ACL-RSI, psychological readiness to return to sport was related to self-reported symptoms and knee function. The aforementioned work also showed that female patients had more negative psychological readiness than male patients. For this reason, they may benefit more from interventions designed to facilitate a smooth transition back to sport [17]. Beyond psychological readiness, it is important to assess pain perception throughout rehabilitation. In this context, subjective pain measurements have been extensively adopted in physical therapy settings and have shown good validity and reliability in various adult populations, albeit limited information is available in athletic populations [18]. Thus, it would be of interest to understand the psychological responses of female athletes to different rehabilitation programs.

Injured athletes can benefit from mental practice, such as hypnosis, meditation, goal setting, self-talk, and imagery [19]. For example, using imagery (a cognitive skill in which the athlete creates or recreates an experience in their mind), the athlete simulates reality by mentally rehearsing a movement, imagining visual, auditory, kinesthetic, olfactory, and even gustatory (taste) cues [20]. Various studies have shown that imagery, in combination with traditional physical therapy, can be effective at reducing psychological distress, such as fear of re-injury and pain perception in first-time anterior cruciate ligament reconstruction patients [21]. However, as traditional imagery often requires silent environment conditions and concentration skills, it is usually replaced by virtual reality-guided imagery (VRMI), which is based on video game-based exercises that contribute to movement facilitation and functional training in ludic ways. This technique promotes movement repetition and improvement of motor skills in different virtual environments [22] and has been shown to improve motor function in post-stroke patients [23]. Beyond the aforementioned techniques, mirror therapy (MT) has received significant attention in the literature. MT is a rehabilitation approach where the reflection (visual input) of a moving non-affected limb gives the illusion of movement in the affected limb. This strategy has been used to treat patients suffering from stroke [24,25], complex regional pain syndrome, and upper extremity surgeries [26]. However, no information is available regarding MT in athletes. 

In summary, both VRMI and MT have been used in isolation to treat patients suffering from different injuries. However, no attempt has been made to understand the effects of combined VRMI and MT added to conventional physical therapy, and no information exists regarding perceptual responses to these rehabilitation strategies in female football players. Thus, this study aimed to examine the effect of MT added to conventional rehabilitation on psychological readiness and perception of pain in ACL-injured female football players.

## 2. Materials and Methods

### 2.1. Participants

A total of 30 ACL-injured female football players competing in the 2nd and 3rd Italian tier who underwent an ACL rehabilitation program from the same clinic participated in this study. Inclusion criteria were an age of >18 years old and a minimum of 3 years competing on the first team at the national level. All procedures followed the guidelines of the Declaration of Helsinki and were in line with established ethical standards in sports sciences [27]. Written informed consent was obtained from all participants.

### 2.2. Design

The participants were recruited throughout a 5-year period, as each player underwent ACL rehabilitation at different times. Players were randomly distributed into an MT group (n = 15; age, mean ± standard deviation [SD] = 33 ± 1.8 years old) and a control group (CON; age = 32 ± 1.6 years old). Both MT and CON interventions lasted 20 weeks. All participants reported pain perception in a VAS before and after the interventions and their psychological readiness in an ACL-RSI after the intervention.

### 2.3. Conventional Physical Therapy

The CON group underwent a 20-week conventional physical therapy program. Training frequency was 5/week with a session length of 60 min. The rehabilitation program was divided into three stages, specifically: Early rehabilitation (1 to 10 weeks): the objectives in this phase are the recovery of the ROM of the injured joint and controlled walking;Advanced mobilization (from week 11 to 16): the objectives are the complete recovery of the ROM and the joint and muscle component;Reconditioning (weeks 17 to 20): the objectives are to improve walking and specific sports exercises for returning to the field.

A certified physiotherapist, based on the acute responses of the patients, adjusted the intensity of the exercise. The conventional physical therapy used in this study comes from the Brotzman and Wilk’s method [28].

### 2.4. Mirror Therapy

The MT group underwent a 20-week conventional physical therapy program plus MT. Training frequency was 5/week with a session length of 60 min, with MT added 3 times/week for 20–25 min after the end of the conventional physical therapy. The MT consisted of the following:Relaxation phase. Typically consists of 3–5 min of relaxation, in which the person is asked to imagine himself in a warm and relaxing place (a beach, a bathtub) and to contract and relax the muscles. Adding a relaxation period should improve concentration, promote a more vivid motor image, and improve attention and performance.Nuclear phase. 10–15 min follow-up in which suggestions are given for imagining internal images related to the use of the affected limb in one or more functional tasks and also through the mirror box.Final Phase. The recording ends with 3 to 5 min of concentrating on what has been done previously.

The teaching tool used is the exercise of mental representation.

Work with exercises in the healthy limb mirror box (work in first person; Table 1).Work with the vision of a video in which, through a background voice, the subject was asked to visualize specific sports movements and gestures (work in first person).

The sets and repetitions vary according to the patient, their health, and pain perception. Specifically, the mental practice includes:-Mirror box (10 reps with healthy limb/10 diseased limb, 3 sets for gradual exercise up to 5 sets recovery every set 1 min). The following presents a series of exercises that were performed during the 1–10-week period:Quadriceps isometric contraction exercises (patient sitting);Flexion/extension of the ankle (patient sitting and standing);Knee flexion/extension (patient sitting and standing);Hip flexion/extension/circling (sitting and standing);Limb adduction/abduction/circling (sitting and standing).-Motor (10 views/1 motor repetition x 5 series increasing to 10) from weeks 11–16:Front/side/back lunge (standing);Single-leg squat (standing);Bipodalic squat (standing);Ascent and descent of steps (standing).-Motor (10 views/1 motor repetition x 5 sets increasing to 10) from weeks 17–20:Stroke (linear, zig zag, 8);Stop (internal, external, with the sole, neck);Passage (internal, neck, external);Jump to hit the head;Guide the ball;Parade.

### 2.5. Subjective Perception of Pain

At the beginning (week 0) and the end of the treatment (week 20), participants were asked to complete the 0 to 10 Likert scale again (anchored to two endpoints labeled “no pain” and “worst pain someone could ever experience”) to report the pain experienced. This scale is a valid, internally consistent, and reliable measure of pain in different populations [18].

### 2.6. Psychological Readiness

At the end of the rehabilitation (week 20), the patients scored their perception of psychological readiness to return to sport after ACL injury and reconstruction surgery on the ACL-RSI scale. The scale was anchored on a 0 to 100 AU and consisted of 12 items, which included 5 related to emotions, 5 to confidence in performance, and 2 to risk appraisal. Emotion-related questions included (1) Are you nervous about playing your sport? (2) Do you find it frustrating to have to consider your knee concerning your sport? (3) Do you feel relaxed about playing your sport? (4) Do you fear reinjuring your knee by playing your sport? (5) Are you afraid of accidentally injuring your knee by playing your sport? Questions related to confidence in performance included (6) Are you confident that your knee will not give way by playing your sport? (7) Are you confident that you could play your sport without concern for your knee? (8) Are you confident about your knee holding up under pressure? (9) Are you confident that you can perform at your previous level of sport participation? (10) Are you confident about your ability to perform well at your sport? Questions related to risk appraisal were (11) Do you think you are likely to reinjure your knee by participating in your sport? and (12) Do thoughts of having to go through surgery and rehabilitation again prevent you from playing your sport? 

This scale was found to have high internal consistency (Cronbach alpha, 0.92), and a fair-to-good predictive ability for a 12-month return-to-sport outcome (area under ROC curve, 0.75) [16].

### 2.7. Statistical Analyses

A Shapiro-Wilks test revealed that ACL-RSI and VAS values were normally distributed within each group at evaluation moments (*p* > 0.05). An independent sample *t*-test was performed to assess between-group differences in post-intervention ACL-RSI. A further independent sample t-test was performed to assume non-significant differences between VAS values before the intervention. A two-way repeated-measures analysis of variance (ANOVA) was used to test for differences in training-induced changes in VAS. The independent variables included one between-subjects factor (training intervention) with two levels (MT and CON) and one within-subject factor (time) with two levels (pre- and post-intervention). To examine the influence of training intervention on the development of our dependent variable, we used this ANOVA to test the null hypothesis of no different change over time between groups (training intervention × time interaction). To qualitatively interpret the magnitude of differences, effect sizes (d), and associated, 95% confidence intervals (95% CI) were classified as small (0.2–0.5), moderate (0.5–0.8), and large (>0.8) [29]. Descriptive statistics were presented as estimated marginal mean ± standard error unless otherwise stated. Statistical significance was set at *p* ≤ 0.05. Data analyses were performed using Statistical Package for Social Science software (IBM SPSS Statistics for Windows, Version 25.0. Armonk, NY, USA).

## 3. Results

A detailed description of participants’ ACL-RSI scores is reported in Table 2. After the intervention, the MT group perceived largely greater psychological readiness (d [95% CIs] = 0.80 [0.53; 0.92]; *p* < 0.01; Figure 1). A detailed description of participants’ VAS scores is reported in Table 3. Both MT and CON groups experienced a large reduction in VAS after the intervention (d [95% CIs] = 0.96 [0.89; 0.98] and 0.88 [0.71; 0.95], respectively; *p* < 0.001; Figure 2). However, a small time × group interaction was observed (d [95% CIs] = 0.25 [0.28; 0.12]; *p* = 0.023).

## 4. Discussion

The main finding of this pilot study was that female football players undergoing MT-based rehabilitation reported a greater perception of psychological readiness compared to those involved in conventional physical therapy. Both groups experienced a similar reduction in pain perception after the rehabilitation. Undergoing ACL rehabilitation sooner after injury, along with a high motivation to return to sports, may impact a player’s return to play after an ACL injury [30].

An injury weakens an athlete’s physical ability, but above all there is the psychological aspect. According to a study, 55% of athletes resume playing sports after surgery, but at a lower level than that practiced before getting hurt [31]. Therefore, it is very important to take care of the rehabilitation adequately. The optimal duration of the mental training sessions was 20–25 min, as it was found that longer sessions could reduce motivation and increase negative effects, such as boredom. Having greater psychological readiness during rehabilitation is an important factor that should not be underestimated concerning returning to sports. As for pain, thanks to mental training, MT footballers have reduced pain perception to almost a minimum, compared to the CON group, which maintained average levels. At the beginning of the project, both the athletes of the MT and CON groups had expressed high values in terms of pain perception on the VAS test, and the constant pain often limited rehabilitation sessions for both groups. Gradually, however, the MT group began to show a more positive and proactive attitude during the sessions. In fact, at the end of the rehabilitation project, when the Sport after Injury scale was administered, it emerged that they had developed adequate self-efficacy to return to sport. Also, in terms of engines, they presented less lameness and fear of returning to the field. This may also be due to a greater regression of pain compared to the control group.

Understanding the relationship between psychological factors and returning to sports is essential because most athletes recover good physical function after surgery, yet, many athletes with good knee function do not return to sports [14]. For example, at the amateur level, an estimated 32% of female football players do not return to sport, and those who do return to competition have a high re-injury risk, particularly to the contralateral limb [32]. Additionally, most athletes who drop out following an injury reported that the main factors preventing them from returning are fear of new trauma and pain [21]. On the other hand, psychological readiness to return to sport and recreation seems to be the factor most strongly associated with returning to preinjury activity [33]. It would be crucial to address these factors during rehabilitation [34,35], and this practice can also be useful for performance strengthening [36,37].

This study has some limitations. First, the sample consists of a small number of participants. Second, the protocol administered did not follow a validation procedure. Finally, pain perception should have been measured at each therapy session; instead, in our case, it was measured only at the beginning and end of treatment. Given these limitations, there is a need to replicate the study to extend beyond the generalization of the results obtained.

## 5. Conclusions

Mirror therapy reported a greater perception of the psychological readiness of the soccer players and a lower perception of pain compared to those who performed conventional therapy. Although both groups experienced a similar reduction in pain perception after rehabilitation, those who performed mirror therapy felt more ready to return to sports. It is important to consider the psychological aspect concerning players, as the fear of new injuries can affect their return to sport. For Mirror Therapy to become a standard module of treatment in sports rehabilitation, there is a need for collaboration between the physical therapist and other staff professionals, such as the mental coach, sports psychologist, osteopath, and orthopedic surgeon. They could define a protocol to improve function and perceived quality of life, not only in the rehabilitation treatment of athletes in the post-acute but also in the chronic phase. The total duration of the treatment could vary from 1 week to 8 weeks. Fifty percent of the treatments have an intervention period of 4 weeks [38], with a frequency ranging from 3 to 5 sessions per week. Each session’s duration could vary from 20 to 90 min.

## Figures and Tables

**Figure 1 jfmk-07-00113-f001:**
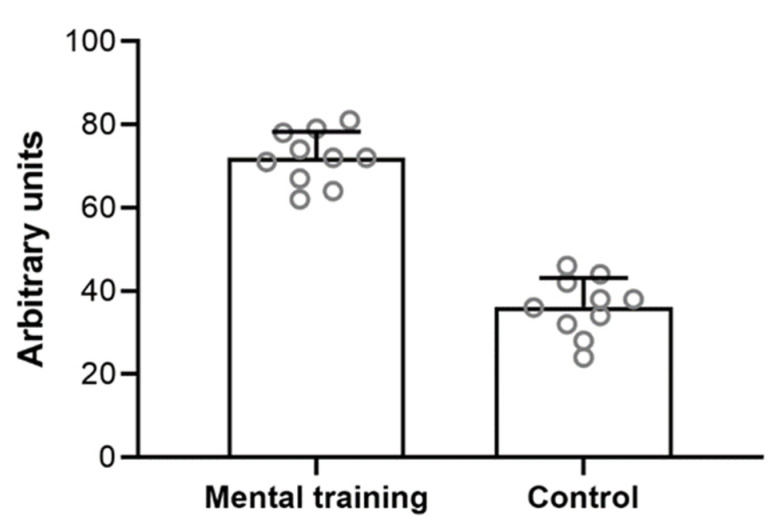
Average perception of psychological readiness after the intervention.

**Figure 2 jfmk-07-00113-f002:**
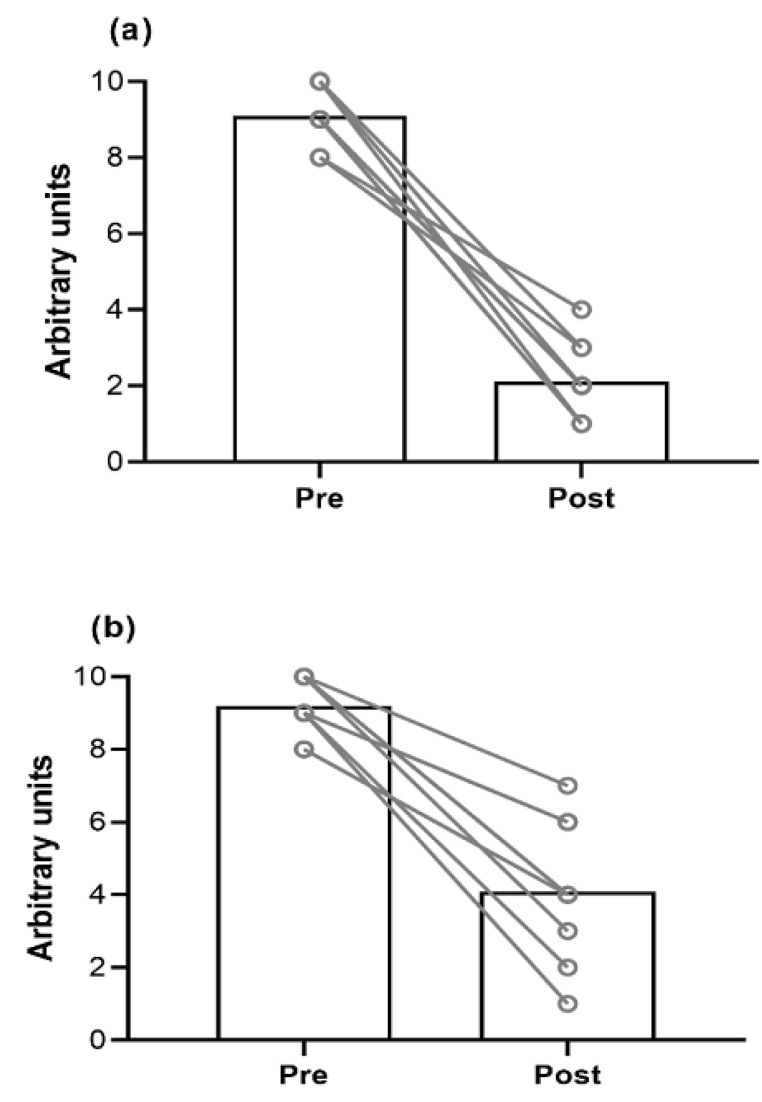
(**a**) Average perception of pain before and after the intervention in the Mirror therapy group. (**b**) Average perception of pain before and after the intervention in the Control group.

**Table 1 jfmk-07-00113-t001:** Mental practice added to conventional physical therapy.

Week 1 to 10	Week 11 to 16	Week 17 to 20
Early rehabilitation	Advanced mobilization	Return to practice/reconditioning
Same as conventional physical therapy	Same as conventional physical therapy	Same as conventional physical therapy
Awareness and motor control	Proprioceptive learning	Technical skills
Joint ROM of the lower limbs and muscle contraction Ankle flexion-extension, knee and hip through the use of the contralateral limb and contraction with Mirror box Awareness of the standing position via video	Video of standing, walking forward, backward, 30° mini squat, side lunges, climbing and descending stairs	Video about the technical gesture: passing, kicking, shooting, parrying, and header

**Table 2 jfmk-07-00113-t002:** Individual Anterior Cruciate Ligament Return to Sport After Injury (ACL-RSI) scores after the rehabilitation.

Participant’ ID	MT Group	CON Group
1	67	42
2	71	24
3	64	28
4	72	38
5	62	44
6	74	38
7	81	32
8	79	46
9	72	36
10	78	34
11	67	35
12	80	36
13	75	40
14	81	36
15	78	35
Mean ± SD	72.00 ± 6.32	36.20 ± 6.96

**Table 3 jfmk-07-00113-t003:** Perception of pain before and after the treatment.

Participant’ ID	MT Group		CON Group	
	Pre	Post	Pre	Post
1	10	2	10	7
2	9	1	10	4
3	10	1	10	3
4	9	2	9	6
5	9	1	9	6
6	9	2	8	4
7	8	3	10	4
8	8	4	8	4
9	10	3	9	2
10	9	2	9	1
11	7	4	7	5
12	8	3	8	6
13	7	5	7	5
14	9	3	8	4
15	10	2	9	5
Mean ± SD	9.1 ± 0.9	2.1 ± 0.9	9.2 ± 1.8	4.1 ± 1.8

## Data Availability

Data available on request from the corresponding author.

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
