# Peer review of "Examining the Effects of Mirror Therapy on Psychological Readiness and Perception of Pain in ACL-Injured Female Football Players"

_jfmk, 2022, doi:10.3390/jfmk7040113_

Round 1
Reviewer 1 Report
The article is very interesting and new for therapists working in sport and rehabilitation area. The study results are reliable to other therapeutic focus. I have great respect to the authors gathering the data over five years. It is a very specific population. The treatment group with N=15 is small. But I can understand, that there are no more for both study groups. N=30.The statistic testing is correct, and the results can be considerate as being significant. I like the discussion. The results are systematically discussed. Step by step. Only point 5 conclusion: " physiotherapists, in collaboration with other professionals, must take care of every aspect of an athlete`s return to sport“ is in my opinion too general. Here I would like a purpose: How often, how much and who should receive the mirror therapies. Who should decide this and who are the other professionals.
So that the mirror therapy can become a standard module of treatment in sport rehabilitation. Maybe this can be mentioned before, so that it is not too long in writing for the conclusion. And then the conclusion become more specific. I like the study very much.
Author Response
Dear Reviewer,
Thank you for reviewing our manuscript. We appreciate it. We have followed your suggestions to improve the manuscript quality, according to our possibilities. Changes have been made to the full text using word tracking to detect changes immediately.
Reviewer 2 Report
The article relates to the effects of mirror therapy on psychological readiness and perception of pain in ACL-injured female football 3 players. After reading, I have few doubts which need explanation before publication.
1. In the abstract there are two group n=15 and in the main text there is said to be 20 participants, 10 in every group. What was the final sample size?
2. I do not like the last sentence in the Conclusions in the abstract. I feel that it is not the main conclusion based on the results. Its some practical application, but not the final result.
3. You mention in the abstract sth. about VRGI, and MT which were added to the conventional therapy. However, in the methods section there is only about MT. How it was provided?
4. Line 219: it relate to which group? What the sentence about the "small time x group ..." relates to?
5. If you add some limitaion, it is also good to explain them somehow or confirm that even if you did sth and you feel it as a limitation, but.....and give some arguments that it was good under some circumstances.
Author Response
Dear Reviewer,
Thank you for reviewing our manuscript. We appreciate it. We have followed your suggestions to improve the manuscript quality, according to our possibilities. Changes have been made to the full text using word tracking to detect changes immediately.
A: Authors
R2: Reviewer 2
R2: In the abstract there are two group n=15 and in the main text there is said to be 20 participants, 10 in every group. What was the final sample size?
A: Thank you for the advice. We have corrected the error in the text. The sample consists of 30 participants, 15 in every group.
R2: I do not like the last sentence in the Conclusions in the abstract. I feel that it is not the main conclusion based on the results. Its some practical application, but not the final result.
A: Thanks for your advice. Based on your report, we decided to remove the sentence from the abstract.
R2: You mention in the abstract sth. about VRGI, and MT which were added to the conventional therapy. However, in the methods section there is only about MT. How it was provided?
A: Thanks for your advice. In the introduction (lines 90-91) we specified that we wanted to understand the effects of combined VRGI and MT treatment added to conventional physical therapy. In addition, in the description of MT treatment, (lines 140-145), we included the use of videos in which, through a background voice, the athletes were asked to visualize specific sports movements and gestures.
R2: Line 219: it relate to which group? What the sentence about the "small time x group ..." relates to?
A: It refers to both groups. To examine the influence of training intervention on the development of our dependent variable, we used two-way repeated-measures ANOVA to test the null hypothesis of no different change over time between groups (training intervention × time interaction) (lines 206-214).
R2: If you add some limitation, it is also good to explain them somehow or confirm that even if you did sth and you feel it as a limitation, but.....and give some arguments that it was good under some circumstances.
A: Thanks for your advice. We have followed your suggestions to improve the manuscript quality, according to our possibilities.
Thanks for your time.
Round 2
Reviewer 2 Report
Thank you for your correcions and the responses.